# *In vitro* and *in vivo* identification of clinically approved drugs that modify *ACE2* expression

Sanju Sinha[1,2,†,‡], Kuoyuan Cheng[1,2,†,‡] (iD), Alejandro A Schäffer[1,‡] (iD), Kenneth Aldape[3,‡], Eyal Schiff[4] & Eytan Ruppin[1,*,‡] (iD)

## Abstract

The COVID-19 pandemic caused by SARS-CoV-2 has is a global health challenge. Angiotensin-converting enzyme 2 (*ACE2*) is the host receptor for SARS-CoV-2 entry. Recent studies have suggested that patients with hypertension and diabetes treated with ACE inhibitors (ACEIs) or angiotensin receptor blockers have a higher risk of COVID-19 infection as these drugs could upregulate *ACE2*, motivating the study of *ACE2* modulation by drugs in current clinical use. Here, we mined published datasets to determine the effects of hundreds of clinically approved drugs on *ACE2* expression. We find that ACEIs are enriched for *ACE2*-upregulating drugs, while antineoplastic agents are enriched for *ACE2*-downregulating drugs. Vorinostat and isotretinoin are the top *ACE2* up/downregulators, respectively, in cell lines. Dexamethasone, a corticosteroid used in treating severe acute respiratory syndrome and COVID-19, significantly upregulates *ACE2* both *in vitro* and *in vivo*. Further top *ACE2* regulators *in vivo* or in primary cells include erlotinib and bleomycin in the lung and vancomycin, cisplatin, and probenecid in the kidney. Our study provides leads for future work studying *ACE2* expression modulators.

**Keywords** angiotensin I-converting enzyme 2; coronavirus disease 2019; dexamethasone; drug-modifying *ACE2* expression; severe acute respiratory syndrome coronavirus 2

**Subject Categories** Microbiology, Virology & Host Pathogen Interaction; Pharmacology & Drug Discovery

**Mol Syst Biol. (2020) 16: e9628**

## Introduction

The ongoing pandemic of coronavirus disease 2019 (COVID-19), caused by the severe acute respiratory syndrome coronavirus 2 (SARS-CoV-2) virus, has plagued so far over 200 countries and has resulted in over 11 million cases and 500,000 deaths since the start of 2020. A key cellular receptor for SARS-CoV-2 entry in humans is angiotensin-converting enzyme 2 (encoded by the gene *ACE2*) (Brielle *et al*, 2020). A recent publication by Fang *et al* (2020) suggested that patients with hypertension (HT) and diabetes mellitus may be at higher risk of having severe COVID-19 disease, as these patients have been reported to express *ACE2* at an increased level. One mechanism is that HT patients are often treated with ACE inhibitors (ACEIs) or angiotensin II type-I receptor blockers (ARBs), which have been previously suggested to increase *ACE2* expression (Ferrario *et al*, 2005; Liang *et al*, 2015; Vuille-dit-Bille *et al*, 2015; Li *et al*, 2017). In contrast to the hypothesis that ACEis or ARBs could be deleterious, an early clinical study of hospitalized COVID-19 patients showed significantly lower mortality among patients taking ACEis or ARBs (Zhang *et al*, 2020).

ACEIs and ARBs are widely used antihypertensive drugs acting on the renin–angiotensin system, a hormone system comprising different variants of angiotensin peptides with important roles in regulating vascular and kidney functions (Dimou *et al*, 2019). ACEIs inhibit the *ACE* gene (but not the *ACE2* gene) whose encoded protein, ACE, converts angiotensin I to angiotensin II. ARBs suppress the blood pressure-increasing effect of angiotensin II by blocking its binding to its receptor (Dimou *et al*, 2019). Distinct from *ACE*, *ACE2* is responsible for the conversion of angiotensin I and angiotensin II into other forms including angiotensin-(1–9) and angiotensin-(1–7), which counteracts the effect of angiotensin II and may paradoxically have protective effects on the lung and on the cardiovascular system (Jiang *et al*, 2014; Paz Ocaranza *et al*, 2020). The possibility that *ACE2* expression may affect either susceptibility to SARS-CoV-2 infection or severity of disease after infection has raised the need to investigate the effects of a variety of prescribed drugs on the modulation of *ACE2* expression.

Addressing this challenge, we aimed to identify drugs whose treatment can alter *ACE2* expression and, assuming that the latter is

1   Cancer Data Science Laboratory (CDSL), National Cancer Institute (NCI), National Institutes of Health (NIH), Bethesda, MD, USA
2   Center for Bioinformatics and Computational Biology, University of Maryland, College Park, MD, USA
3   Laboratory of Pathology, National Cancer Institute (NCI), National Institutes of Health (NIH), Bethesda, MD, USA
4   Department of Obstetrics, Gynecology and Reproductive Sciences, Chaim Sheba Medical Center, Sackler Faculty of Medicine, Tel-Aviv University, Ramat Gan, Tel-Aviv, Israel
    *Corresponding author. Tel: +1 240 858 3169; E-mail: eyruppin@gmail.com
    †These authors contributed equally to this work as first authors
    ‡This article has been contributed to by US Government employees and their work is in the public domain in the USA

an important determinant, possibly increase or decrease the infection risk of COVID-19. To this end, we analyzed the Connectivity Map (CMAP) dataset that provides transcriptomic data of a collection of cell lines treated with approximately 20,000 small molecules (Subramanian et al, 2017). Utilizing this resource, we mined the ACE2 expression fold change (logFC) after each drug treatment, to identify clinically approved drugs that result in strong upregulation or downregulation of ACE2 expression in this data. We additionally mined the Gene Expression Omnibus (GEO) and the Gene Tissue Expression Consortium resource (GTEx) datasets aiming to identify which of the emerging in vitro findings may have further in vivo support.

## Results

We first focused on antihypertensive (anti-HT) drug treatment data from the CMAP dataset to study the suggestion of Fang et al (2020) that certain anti-HT drugs may affect ACE2 expression. Among the available cell types from CMAP, we focused on carcinoma cell lines, since they are of epithelial origin and may bear more resemblance to airway epithelium, a major site of viral entry. We identified 48 clinically approved anti-HT drugs that were tested on the same four carcinoma cell lines for up to 24 h in CMAP and computed the ACE2 expression changes after their treatment averaged across the cell lines (Materials and Methods; the cell lines are A549, MCF7, PC3, and VCAP, selected because of the data available for a high number of drugs tested on all these cells, see Appendix Note S1, Appendix Fig S1 and Table EV1A for details).

Individually, no widely prescribed anti-HT drug was found to increase ACE2 expression significantly in these experiments, but methyldopa (an alpha-2 adrenergic receptor agonist) and molsidomine (a vasodilator) do significantly decrease ACE2 expression (Fig 1A, logFC = −0.605 and −0.290, P = 0.002 and 0.005, respectively; adjusted P = 0.11 for both; Table EV1B). When the individual drug results are aggregated to identify the effects of major classes of anti-HT drugs, we find that ACEIs, but not ARBs, tend to upregulate ACE2 expression (Fig 1B, Gene set enrichment analysis (GSEA) method P = 0.026, adjusted P < 0.1; Table EV1C). Anti-adrenergics other than alpha/beta-blockers tend to downregulate ACE2 (Fig 1B, GSEA P = 0.032, adjusted P < 0.1; Table EV1C). Notably, we find that calcium channel blockers (CCBs) do not significantly change ACE2 expression, consistent with the fact that they do not act on the renin–angiotensin system. This finding provides preliminary in vitro support for the suggestion of Fang et al (2020) that CCBs may be considered as an alternative to ACEIs and is further supported by results from a large study cohort where hypertensive patients treated with CCBs (amlodipine and nifedipine) had no increase in urinary ACE2 levels compared with untreated controls (Furuhashi et al, 2015). A similar analysis for the 13 approved antidiabetic drugs in the CMAP dataset that were tested on the same four carcinoma cell lines did not identify any individual or class of drugs that significantly altered ACE2 expression, partly due to the small number of drugs in this class (Table EV1D and E).

We then turned to analyze a broad set of 672 clinically approved drugs, each of which was tested on the same four carcinoma cell lines as described above in the CMAP dataset, to identify the top drugs that upregulate or downregulate ACE2 expression (full results

in Table EV1F). The top upregulators are vorinostat and panobinostat, both of which are anticancer histone deacetylase (HDAC) inhibitors (Fig 1C and D, logFC = 0.321 and 0.457, adjusted P = 2.77e−18 and 7.70e−3, respectively). They have previously been shown to have antifibrotic effects and can reduce the risk of acute respiratory deterioration (Lyu et al, 2019; Maher & Strek, 2019). The top ACE2 downregulator is isotretinoin, a vitamin A derivative with suspected respiratory side effects (Gorpelioglu et al, 2010) (Fig 1C and D, logFC = −0.478, adjusted P = 0.036). Notably, we identified six clinically approved drugs in CMAP that are currently being investigated in clinical trials (www.clinicaltrials.gov) for COVID-19 (chloroquine, thalidomide, methylprednisolone, losartan, lopinavir, and ritonavir), none of which was found to alter ACE2 expression significantly (P > 0.1, Table EV1F).

Analyzing the expression results of these 672 drugs in an aggregated manner, we performed a GSEA based on the drug-induced ACE2 expression fold changes and identified several classes of drugs based on mechanism of action (MOA) that are significantly enriched for up or downregulation of ACE2 expression, using the MOA annotation from the Drug Repurposing Hub (Corsello et al, 2017) (Fig 1E; Materials and Methods; full result in Table EV1G). Top classes of ACE2-upregulating drugs are HDAC inhibitors (GSEA P = 0.003, adjusted P = 0.02) and dopamine receptor antagonists (GSEA P = 0.007, adjusted P = 0.02). We further extended this analysis to 989 clinically approved drugs tested on a total of 28 CMAP cell lines. The 28 cell lines include 16 cancer and 12 normal cells (this list is provided in Table EV2A with details including primary site, subtype, and donor demographics), where each drug may have been tested on a different subset of cells (Materials and Methods). We examined the enrichment of ACE2-modulating drugs in the different WHO Anatomical Therapeutic Chemical (ATC) indication categories (World Health Organization, 2006) (Fig 1F; Materials and Methods; full results in Table EV2C, based on ACE differential expression result in Table EV2B). We find that the class of drugs targeting the renin–angiotensin system is enriched for upregulators of ACE2 (GSEA P = 0.044), while antineoplastic agents, and in particular protein kinase inhibitors, are enriched for downregulators of ACE2 expression (GSEA P = 0.004), although these classes do not achieve significance after FDR correction (Table EV2C). Specific protein kinase inhibitors were also previously shown to inhibit MERS-CoV and SARS-CoV in vitro (Dyall et al, 2014).

The analyses described above were performed by aggregating the drug-induced expression changes across cell types (Materials and Methods). We next analyzed CMAP data of additional relevant cell types separately by their tissue of origin to investigate potential tissue-specific effects. We focused on the lung, kidney, liver, central nervous system (CNS), and intestine (Materials and Methods), which represent tissues that can be affected by SARS-CoV-2 (Zaim et al, 2020). For each of these tissues, we were only able to find one (or two, for lung) cell type where a reasonable number (> 100) of clinically approved drugs have been tested at the 24-h time point (details in Table EV2D). The cells identified from kidney, liver, and CNS were non-cancerous or primary cells (HA1E, PHH, and NPC cells, respectively), which could be more relevant for our investigation than the cancer cell lines from CMAP. As expected, the drug-induced ACE2 expression changes exhibit mostly weak correlations across cells from different tissue types, with Spearman's correlation coefficients between the log fold changes of pairs of cells ranging

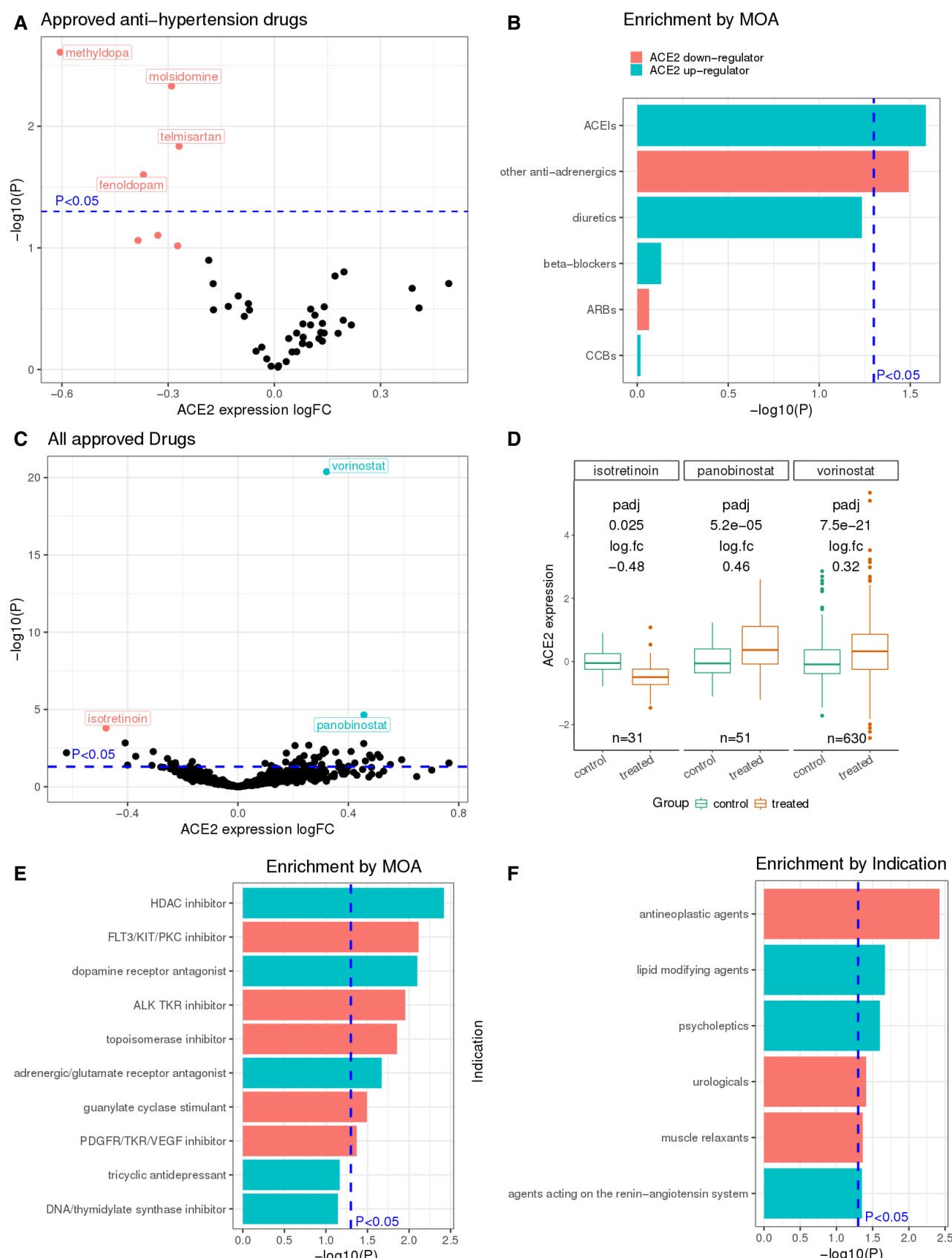

**Figure 1.**

◄

**Figure 1.   The landscape of *ACE2* expression levels alterations in response to drug treatments across cell lines in CMAP (Subramanian *et al*, 2017).**

A–F   Differential expression analysis for *ACE2* expression changes in treated vs control samples was performed with the level 3 CMAP data of 24-h response values using limma (Ritchie *et al*, 2015; Materials and Methods). Volcano plots showing the log fold change (*x*-axis) and uncorrected negative $\log_{10}P$ value (*y*-axis) of *ACE2* expression changes are given for (A) 48 antihypertensive drugs and (C) 672 clinically approved drugs that are each tested on four carcinoma cell lines from CMAP (Subramanian *et al*, 2017). The *ACE2* expression levels in control and drug-treated groups for the top significant drugs (vorinostat, panobinostat, and isotretinoin) among the 672 clinically approved drugs are shown with box plots in (D), the *ACE2* expression log fold change values and adjusted *P* values from limma, and the number of samples per group are labeled for each drug. In the box plot of panel (D), the center line, box edges, and whiskers denote the median, interquartile range, and the rest of the distribution in respective order, except for points that were determined to be outliers using a method that is a function of the interquartile range, as in standard box plots. The enrichment of positive/negative *ACE2* expression regulators in different drug classes based on their mechanism of action (MOA) was tested with the GSEA method as implemented in the R package fgsea (preprint: Korotkevich *et al*, 2019; Materials and Methods), and the enrichment significance (negative $\log_{10}P$ values) is shown in bar plots in (B) and (E), for the analysis on the 48 antihypertensive drugs and the 672 clinically approved drugs, respectively. We further extended the analysis to a larger set of 989 clinically approved drugs tested on a total of 28 cell lines (16 cancer and 12 normal cells; this list is provided in Table EV2A with details including primary site, subtype and donor demographics) where each drug may have been tested on a different subset of cells (Materials and Methods) and performed enrichment analysis using the WHO ATC drug classification data (World Health Organization, 2006), and top enriched drug classes are shown in (F). The horizontal and the vertical dashed lines denote a *P* value of 0.05. Drug indication and ATC classification annotations are obtained from the DrugBank database (Wishart *et al*, 2018). All *P* values are computed from differential expression analysis using limma (Ritchie *et al*, 2015) (Materials and Methods). ACEIs, angiotensin-converting enzyme inhibitors; ARBs, angiotensin II type-I receptor blockers; CCBs, calcium channel blockers; ERAs, endothelin receptor antagonists; TKR, tyrosine kinase receptor.

from −0.07 to 0.2 (Fig EV1A; Table EV2E). Nevertheless, we observed a consistent but insignificant trend that ACEIs tend to upregulate *ACE2* expression across the three normal cell types from kidney, liver, and CNS (Fig EV1B). Concordant with the findings above from the four carcinoma cell lines in CMAP, antineoplastic agents as a group were found to be enriched for drugs downregulating *ACE2* in the normal NPC cells from CNS (GSEA adjusted *P* = 0.01, Fig EV1C, Table EV2F).

Additionally, we also analyzed CMAP drug treatment data beyond the 24-h time point used above. We identified 14 clinically approved drugs with such data available, which were tested on either the 293T or VCAP cell lines for 48 h (details and results in Table EV2G). In these data, vemurafenib, fluphenazine, and afatinib were found to significantly upregulate *ACE2* expression (adjusted *P* < 0.1), and imatinib significantly downregulated *ACE2* (adjusted *P* = 0.001). Treatment data at the 6- and 24-h time points in the same cell line were available for vemurafenib and fluphenazine; for both drugs, we observed a trend of time-dependent increase in the level of *ACE2* upregulation (Fig EV2), suggesting that these drugs may modulate *ACE2* expression during prolonged treatment.

To extend our analysis beyond the CMAP dataset, we mined the GEO database for gene expression data of drug treatments with matched controls in lung and kidney tissue or cells (Materials and Methods). We collected a total of 74 relevant lung datasets involving 42 unique clinically approved drugs, among which 27 datasets (covering 21 drugs) were composed of non-cancerous samples including primary bronchial epithelial cells and *in vivo* samples from human and rodents (Table EV3A). Similarly, for kidney, 35 datasets for 29 drugs (including 23 drugs in 28 non-cancer datasets involving *in vivo* samples) were identified (Table EV3B). The drug-induced *ACE2* differential expression results (Materials and Methods) for the lung and kidney datasets are summarized in Fig 2A and C, respectively. The top significant drugs identified from the more relevant non-cancer datasets are visualized in Fig 2B and D. For lung, the most significant drug is dexamethasone, which upregulates *ACE2* in datasets of both normal and *Pneumocystis*-infected mice lung tissue (logFC = 0.97 and 0.36, adjusted *P* = 0.001 and 0.027, respectively, Fig 2B). Dexamethasone also increased *ACE2* expression in our analysis of four carcinoma cell lines from the CMAP dataset (logFC = 0.18, *P* = 0.006, Table EV1F). Notwithstanding, one

should note that recent studies have reported that dexamethasone treatment of hospitalized COVID-19 patients has beneficial effects, but this might be due to its immune-modulatory actions (preprint: Horby *et al*, 2020). Another top identified drug is the epidermal growth factor receptor (EGFR) inhibitor erlotinib, which is found to upregulate *ACE2* in a dataset of human primary bronchial epithelial cells (logFC = 1.04, adjusted *P* = 2.95E-5, Fig 2B), a relevant cell type suggested to interact with the SARS-CoV-2 virus (Mason, 2020). In the CMAP analysis, we observed a non-significant trend of *ACE2* upregulation at 24 h by erlotinib (logFC = 0.05, adjusted *P* > 0.1, Table EV1F). Interestingly, erlotinib has been reported to inhibit the endocytosis and intracellular trafficking of multiple viruses including hepatitis C, dengue, and Ebola, exerting broad-spectrum antiviral effects (Bekerman *et al*, 2017). The chemotherapeutic drug bleomycin is a significant *ACE2* downregulator identified in a dataset of rat lung tissue (logFC = −0.17, adjusted *P* = 0.003, Fig 2B), in accordance with an earlier report that bleomycin decreases *ACE2* protein level in alveolar epithelial cells (Uhal & Abdul-Hafez, 2010).

Among the most significant candidates in the kidney cell analysis (summarized in Fig 2C), we again focused on non-cancer datasets and observed that the chemotherapy drug cisplatin upregulated *ACE2* in mice kidney samples, while it downregulated *ACE2* in the renal cortex of rat (logFC = 0.29 and −1.16, adjusted *P* = 8.06E−3 and 2.36E−5, respectively, Fig 2D), suggesting a cell type and possibly species-specific effect. Vancomycin, another top identified drug, is a glycopeptide antibiotic that increases *ACE2* expression in mice kidney samples from two independent datasets (logFC = 0.89 and 0.93, adjusted *P* = 0.04 for both). Glycopeptide antibiotics and its derivatives have been previously shown to block MERS and SARS cell entry (Zhou *et al*, 2016). Probenecid, a drug for treating gout, was found to decrease *ACE2* expression in a renal cortical cell line (logFC = −0.61, adjusted *P* = 0.001, Fig 2D); this drug has been proposed to be repurposed for anti-influenza therapy (Perwitasari *et al*, 2013). Other significant drugs arising from the analysis of cancer datasets of lung and kidney from GEO are shown in Fig EV3 with additional information in Appendix Note S2.

Finally, we mined gene expression in normal human lung tissue from the GTEx dataset to extend our *in vitro* findings

                                                                    

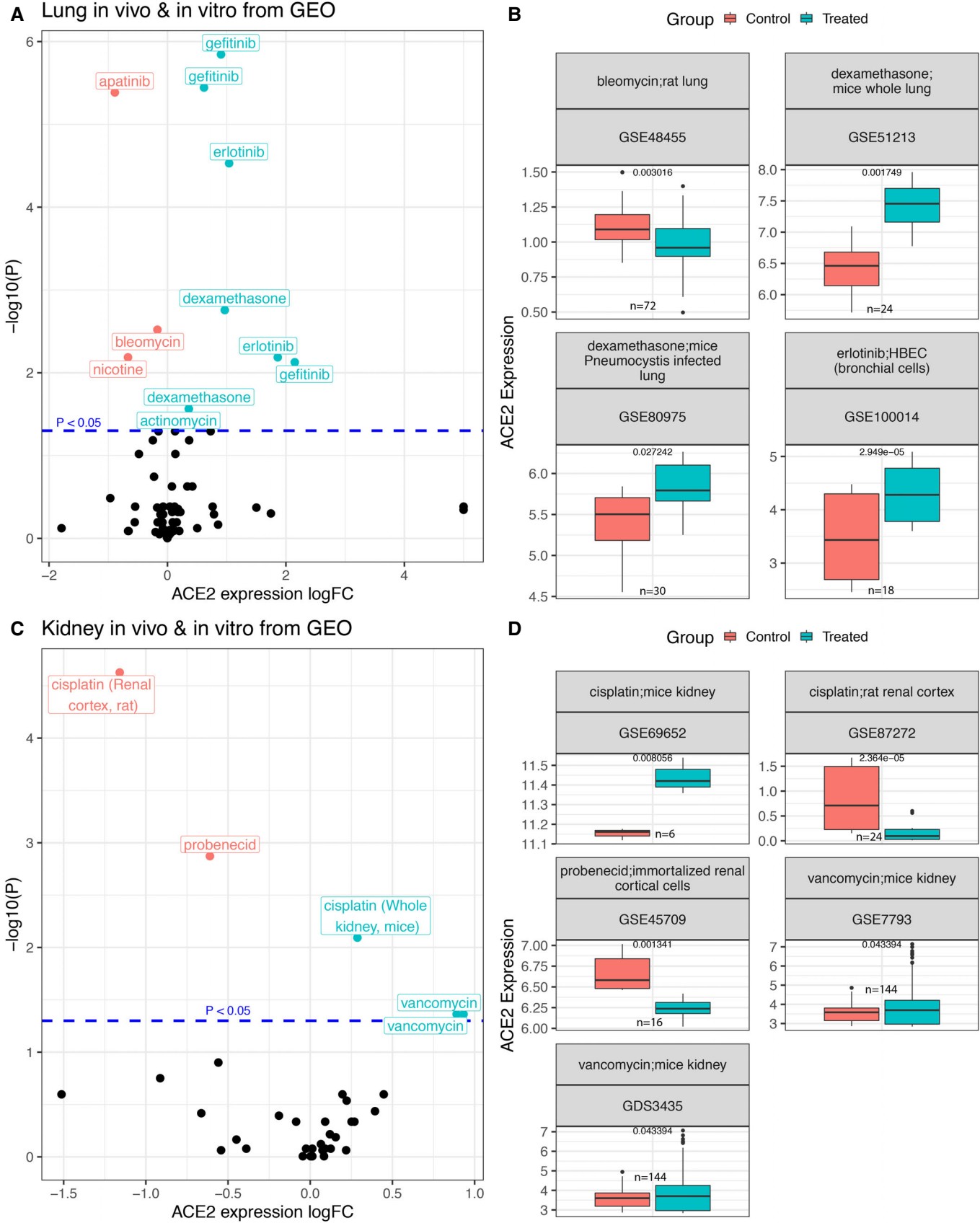

Figure 2.

**Figure 2. Drug-induced *ACE2* differential expression in lung and kidney-derived cells or tissue types from the GEO database.**

A–D  This fig summarizes the differential expression analysis results for *ACE2* upon treatment by clinically approved drugs in samples of lung and kidney datasets mined from the GEO database, spanning both cancer and non-cancer datasets (Materials and Methods). Volcano plots showing the log fold change (*x*-axis) and uncorrected negative $\log_{10}P$ value (*y*-axis) of *ACE2* expression changes are displayed in (A) for 74 datasets of cells or tissue samples from lung, involving 42 clinically approved drugs, and in (C) for 35 kidney datasets involving 29 clinically approved drugs. Among these, we focused on the top significant drugs that modulate *ACE2* expression in non-cancerous cells or tissue samples from lung and kidney, which are shown in (B) and (D), respectively, where the *ACE2* expression (*y*-axis) difference between control and treated groups (*x*-axis) are shown with box plots. In the title of each box plot, the GEO identifier of the respective studies and the corresponding drug name and the sample type are provided. Different datasets involving experiments using the same drug were analyzed and presented separately, since the sample types can be different across datasets. All the *P* values are computed from differential expression analysis using limma (Ritchie *et al*, 2015) (Materials and Methods). In the box plots of panels (B) and (D), the center line, box edges, and whiskers denote the median, interquartile range, and the rest of the distribution in the respective order, except for points that were determined to be outliers using a method that is a function of the interquartile range, as in standard box plots. HBEC, human primary bronchial epithelial cells.

(Lonsdale *et al*, 2013). To this end, we identified genes whose expression is positively or negatively associated with *ACE2* expression, and then examined whether targets of certain classes of drugs are enriched among those genes. This analysis is based on the notion that the effect of drug treatment, which usually acts on the protein level, is functionally similar to the effect of downregulating the drug target expression on the mRNA level. Consistent with our earlier findings in CMAP, we find that the targets of ACEIs are enriched for genes negatively associated with *ACE2* expression in normal human lung tissue (Fisher's test adjusted $P = 8.58\text{E}{-}4$, Materials and Methods), while the targets of antineoplastic drugs are enriched for genes positively correlated with *ACE2* expression (adjusted $P = 2.24\text{E}{-}4$). Additionally, the latter association for antineoplastic drug targets is also observed in a single-cell RNA-seq dataset of normal lung tissue (adjusted $P = 7\text{E}{-}3$) (Vieira Braga *et al*, 2019).

## Discussion

To date, solid evidence concerning whether *ACE2* expression can alter the risk of COVID-19 infection is lacking. The role of ACEIs/ARBs in modulating the clinical course of viral pneumonia and COVID-19 infection is also under debate (Diaz, 2020; Esler & Esler, 2020; Gurwitz, 2020; Zhang *et al*, 2020; Zheng *et al*, 2020). If *ACE2* expression does influence either susceptibility or disease course, it is important to chart the landscape of how commonly used therapeutic drugs affect *ACE2* expression. Addressing this challenge, we performed a systematic *in vitro* analysis of the CMAP cell line data that shows that ACEIs, although not most other antihypertensives, are enriched for upregulators of *ACE2* expression (Fig 1B). Extending the scope of our analyses, we identified additional clinically approved drugs and drug categories that affect *ACE2* expression *in vitro* (Fig 1C–E). Analyzing the additional but limited amount of CMAP data for cells from different tissue types and for drug treatment for longer durations up to 48 h, we find that the drug-induced *ACE2* changes can be tissue-specific and time-dependent (Figs EV1 and EV2), and for some drugs, their effect on *ACE2* can become stronger upon prolonged treatment (Fig EV2). We successfully corroborated some of the drugs' effects identified in CMAP, mining *in vitro and in vivo* gene expression data of the lung and kidney from the GEO database. The analysis of GEO data from primary or immortalized cells also resulted in additional candidate *ACE2* regulators, including a corticosteroid (dexamethasone), antineoplastic drugs (erlotinib, bleomycin and cisplatin tested in non-cancerous

samples), an antibiotic (vancomycin), and an uricosuric agent (probenecid) to be further investigated in experimental studies assessing their effects on *ACE2* expression, which may be relevant to SARS-CoV-2 infection and pathogenesis.

## Materials and Methods

### The CMAP data

The level 3 data of the Connectivity Map (CMAP) dataset (Subramanian *et al*, 2017) were downloaded from the GEO database (GSE92742 and GSE70138). We used the DrugBank database (Wishart *et al*, 2018) to identify all drugs that are clinically approved in the United States, Canada, or the EU and then matched the drugs to the compounds in CMAP by drug generic names. Starting from this subset of CMAP data for only the clinically approved drugs, we first aimed to choose a reasonably large set of drugs such that each of the drugs in the set were tested on the same set of cell types (also at the same concentration and duration of treatment) for consistency. Due to incomplete data in CMAP, there exists a trade-off between the number of cell types and the number of drugs to include (Appendix Note S1 and Fig S1). Balancing this trade-off, we selected a set of 672 clinically approved drugs, each of which had been tested on the same four carcinoma cell lines (A549, MCF7, PC3, and VCAP, Table EV1; the epithelium-derived carcinoma cell may also better resemble the airway epithelial cells relevant for COVID-19 as explained in the main text) for 24 h at a concentration of 10 μM. The chosen time point and concentration represent the most frequent treatment condition present in the dataset. This subset of 672 approved drugs includes 48 antihypertensive and 13 antidiabetic drugs, which we used for the first part of our analysis. In the later parts of our analyses, (i) trading biological homogeneity for higher coverage, we included drugs that each can be tested on a different number and types of cells, resulting in 989 clinically approved drugs tested on a total of 28 cell types, which we used for the analysis of enrichment of drug classes by WHO ATC indications; (ii) we filtered CMAP data separately for each tissue type that can be affected by COVID-19, including lung, kidney, liver, central nervous system, and intestine; for each of these tissues, only one (two for lung) cell type was found with data for treatment by > 100 approved drugs (details in Table EV1), and we used these data subsets for the analysis of tissue type-specific *ACE2* response to drugs; (iii) we selected CMAP data with drug treatment for 48 h, resulting in 14 approved drugs each tested on either 293T or VCAP

cell line, and we used this subset of data to investigate the time dependency of drug-induced *ACE2* changes.

### Identification of *ACE2* modulators from the CMAP dataset

Using each of the subsets of CMAP data as described above, we selected the expression data of only the "landmark" and "BING" (best-inferred) genes for the population controls and the drug-treated samples. The landmark/BING genes and population controls are previously described in the CMAP publication (Subramanian *et al*, 2017); *ACE2* is not a landmark gene but is a best-inferred gene. For each drug, differential expression (DE) analysis of drug-treated samples vs population controls was performed using limma (Ritchie *et al*, 2015) taking advantage of the data across all landmark/BING genes, and finally, the DE results for *ACE2* were selected. In cases where each drug was tested on more than one cell type, for example in the first part of our analysis where all drugs were tested on the same four carcinoma cell lines (described above), cell type was included as a covariate in the limma linear model (Ritchie *et al*, 2015), i.e., the results represent averaged DE across the cell types.

### Analysis of drug classes enrichment in *ACE2* modulators

With the drug-induced *ACE2* differential expression results across the selected clinically approved drugs as described above, we tested for the enrichment of different classes of drugs for positive/negative *ACE2* modulators with the GSEA method as implemented in the R package fgsea (preprint: Korotkevich *et al*, 2019). Specifically, the drug-induced *ACE2* expression log fold change values were ranked, and the GSEA method was applied to the ranked list with "gene sets" for GSEA being the sets of drugs of each class. The drug classes were based on mechanisms of action (MOA) and indication in our two respective analysis described in the main text, where the MOA annotation was obtained from the Drug Repurposing Hub (Corsello *et al*, 2017), and for drug indication, we used the WHO ATC classification (World Health Organization, 2006) obtained from the DrugBank database (Wishart *et al*, 2018).

### Identification of *ACE2* modulators using the GEO database

We systematically mined the GEO database for gene expression data of drug treatment with matched controls in lung and kidney cells/tissues combining programmatic search and manual curation. Specifically, we downloaded an SQL database of GEO metadata (timestamp: May 10, 2020) using the R package GEOmetadb (Zhu *et al*, 2008) and used an in-house script (see Data availability) to query the metadata for GEO datasets and data series whose summary, description, or study design information contain the generic names of any of the clinically approved drugs (identified from DrugBank, as described above). We then further filtered the resulting datasets to find keywords such as "lung/pneumo", "bronchial/bronchus", "kidney/renal/nephr", and "treat" (see our script for details; Data availability) to obtain a smaller set of several hundreds of candidate GEO studies. These were then manually selected for relevant studies with drug-treated and control expression profiling. We were careful to exclude studies that compare drug responses in two groups differed by genotype at a single locus, such

as between a wild-type cell line and an isogenic mutant cell line. For each selected relevant study, the gene expression and phenotypic data were downloaded from GEO, the control group and the treated group(s) were manually labeled, and differential expression analysis between treated and control groups was performed with limma (Ritchie *et al*, 2015). If a single GEO study contains experiments of treatment by different drugs, the analysis was then performed individually for each drug. Similarly, for cases where the same drug was tested on different sample types in a study (e.g., a cancer cell line and a normal cell), each cell line was analyzed separately. In cases where multiple time points of drug treatment are available, the drug treatment time was controlled for as a covariate in the limma linear model. Details on those finally selected studies are given in Table EV3A and B.

### Co-expression analysis of drug targets and *ACE2* expression

We computed the association between the expression of *ACE2* and each gene in the GTEx human lung tissue (Lonsdale *et al*, 2013) and mapped the genes to clinically approved drugs that target (i.e., inhibit) them using drug target annotation from DrugBank (Wishart *et al*, 2018). We then identified the two sets of drugs that target only genes with significant positive or negative correlations with *ACE2*, respectively (i.e., drugs targeting multiple genes with mixed directions of correlation with *ACE2* are excluded). The enrichment of these two sets of drugs in the WHO ATC classes of drug indication (World Health Organization, 2006) was analyzed using Fisher's test. Similarly, we computed the correlation of each gene with *ACE2* expression in a single-cell RNA-seq dataset comprising 10,360 cells from upper and lower airways and lung parenchyma in healthy lungs (GSE130148) (Vieira Braga *et al*, 2019) and mapped each gene to the corresponding targeting drugs. For this list, we performed GSEA (preprint: Korotkevich *et al*, 2019) to test for enrichment in the WHO ATC classes of drug indication (World Health Organization, 2006).

*P* values in this study were adjusted with the Benjamini–Hochberg method.

## Data availability

All the data analyzed in this study are from published studies or publicly accessible datasets. The CMAP data were downloaded from the GEO database (GSE92742 [http://www.ncbi.nlm.nih.gov/geo/query/acc.cgi?acc = GSE92742] and GSE70138 [http://www.ncbi.nlm.nih.gov/geo/query/acc.cgi?acc = GSE70138]). The identifiers of the other GEO datasets we curated and analyzed (those containing drug treatment data of lung and kidney-derived cell or tissue samples) can be found in Table EV3. The code used for the analysis can be obtained from https://github.com/ruppinlab/ACE2_modulating_drugs.

**Expanded View** for this article is available online.

### Acknowledgements

This research was supported in part by the Intramural Research Program of the National Institutes of Health, NCI, and used the computational resources of the NIH HPC Biowulf cluster (http://hpc.nih.gov). We acknowledge and thank the National Cancer Institute for providing financial and infrastructural

support. S.S and K.C. are supported by the NCI-UMD Partnership for Integrative Cancer Research Program. We thank Dr. Fiorella Schischlik, Sanna Madan, and Dr. Bríd Ryan for constructive comments on this study.

## Author contributions

ER conceived and supervised the study. SS, KC, and ER designed and developed the methodology. KC and SS acquired and analyzed the data. KC, SS, ER, AAS, KA, and ES wrote, reviewed, and revised the manuscript.

## Conflict of interest

The authors declare that they have no conflict of interest.

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
