## [Review Process File · Molecular Systems Biology]

In vitro and *in vivo* identification of clinically-approved drugs that modify ACE2 expression

Sanju Sinha, Kuoyuan Cheng, Alejandro Schäffer, Kenneth Aldape, Eyal Schiff, and Eytan Ruppin
DOI: [10.15252/msb.20209628](https://doi.org/10.15252/msb.20209628)

Corresponding author(s): Eytan Ruppin (eyruppin@gmail.com)

Review Timeline:

Submission Date:	10th Apr 20
Editorial Decision:	14th May 20
Revision Received:	14th Jun 20
Editorial Decision:	1st Jul 20
Revision Received:	6th Jul 20
Accepted:	7th Jul 20

Editor: Maria Polychronidou

Transaction Report:

Thank you again for submitting your work to Molecular Systems Biology. We have now heard back from the three referees who agreed to evaluate your study. Overall, the reviewers think that despite the modest conceptual novelty of the approach, the presented findings seem relevant given that analysing the effects of drugs on ACE2 is a timely topic. They raise however a series of concerns, which we would ask you to address in a major revision.

As you will see below, the reviewers make constructive suggestions on how to improve the study. Without repeating all the points raised, some of the more fundamental issues are the following:

- Reviewer #1 mentions that assessing the effect of the drugs on ACE2 in non-tumour bronchial epithelial cell lines seems like a better choice compared to cancer cell lines.
- Reviewers #2 and #3 point out that the second and more relevant part of the study, reporting an unbiased analysis of all approved compounds needs to be presented more thoroughly and the related conclusions need to be better supported. Reviewer #3 recommends including additional data (e.g. from different exposure times) for the drugs proposed to be promising for further analyses.
- The methodology needs to be described in better detail in order to be better accessible to the readers.
- Importantly, along the lines of the comment by reviewer #3, we would ask you to make the code available on GitHub and provide the link in the Data Availability section.

All other issues raised by the reviewers should be convincingly addressed. Please let me know in case you would like to discuss any of the issues raised. As this is a topic of active interest, we would ask you to make sure that the latest literature is properly referenced in the manuscript.

.

REFEREE REPORTS

Reviewer #1:

The study on ACE2 expression by a wide variety of clinically approved drugs is an important contribution to better understand the consequences such drugs might have for COVID-19 patients. However, there are some parts of the study that need to be explained in more details. The use of 4 cancer cell lines to test the effect of drugs on ACE2 expression seems not the best approach, as cancer cells are inhibited by ACE2. It would have been better to use some of the non-tumour bronchial epithelial cell lines, but may be the authors can provide a bit more information why they had chosen these cells.

Overall, the study has merits to be published after minor corrections.

1. The correspondence by Fang et al hypothesized that hypertension and diabetes develop severe COVID-19 because the two diseases have been reported to express ACE2 at an increased level. In addition, these patients are often treated with drugs that further increase ACE2. Please correct the statement.
2. Page 2: References # 3 is incorrectly cited. They did not provide data showing that any of the drugs increased ACE2. Replace by e.g. Liang Y, et al The use of cancer cell Kidney Blood Press Res. 2015;40(2):101-10. doi: 10.1159/000368486. Epub 2015 Mar 13; or by Vuille-dit-Bille RN, et al. Amino Acids. 2015 Apr;47(4):693-705. doi: 10.1007/s00726-014-1889-6.
3. Why did the authors use four cancer cell lines to determine the effect of antihypertensive drugs on ACE2. It would have been advisable to use cell lines such as: BEAS2 or NuLi or NCI H441 for these experiments. There are also a number of commercial available primary human bronchial epithelial cells available. The problem with this approach is that ACE2 is inhibiting many epithelium derived cancer types (Cheng Q, et al. Oncol Rep. 2016 Sep;36(3):1403-10. doi: 10.3892/or.2016.4967; Zong H, et al. Tumour Biol. 2015 Jul;36(7):5171-7. doi: 10.1007/s13277-015-3171-2; Yu X, et al. Int J Mol Med. 2018 Jan;41(1):409-420. doi: 10.3892/ijmm.2017.3252.)
4. On page 6 the authors state that the drugs effect on ACE2 expression was verified 28 cell lines. Please provide more information on the cell type and of their origin. It is unclear if these 28 cell lines result from the different studies or if the authors were performing these assays. If the latter is true, than they should provide more details of the cells.

Reviewer #2:

The authors present results from a quick and simple computational analysis of public available repositories of transcriptional data to the aim of identifying modulators of the expression of ACE2: a receptor required for SARS-CoV-2 infection in humans.

To this aim, the authors queried the connectivity map (cMap) database of transcriptional responses to drug treatment in immortalised human cancer cell lines, initially focusing on anti-hypertensive and anti-diabetic drugs, i.e. ACE inhibitors (ACEi) and angiotensin II type1 receptor blockers (ARBs). Consistently with previous findings (PMID: 20838579) the authors found that ACE inhibitors increases ACE2 expression but no ARBs was associated with differential expression of ACE2.

Furthermore the authors extended this analysis to all approved compounds with data available in the cMap and performed an enrichment analysis of drug Mode-of-action and therapeutic

applications among the compounds exerting an effect on ACE2 expression.

Finally, the authors confirmed some of their hits using an independent dataset including drug transcriptional responses in vivo and attempted confirmatory a gene co-expression analysis using public data from normal lung tissue.

The recent COVID-19 outbreak and the need for findings and results that might help discovering/repositioning drugs able to reduce infections rate and symptoms make the subject of this report timely and important. However the analytical approach presented in this report is not novel, there is no new data presented nor final strong statements or guidelines for experimental followups, with just percentages of drugs with confirmatory in-vivo results reported, and not even stating the names of the most promising hits.

In addition, the first part of the paper focuses too much on confirmatory results and previously reported findings related to ant hypertensive and anti-diabetic drugs, whereas the more interesting unbiased and comprehensive analysis of all approved compounds is presented quite poorly.

Particularly the following points should be addressed, in my opinion:

- * The methods section should be generally extended, for example is not clear whether the differential ACE2 expression analysis for a given drug was performed considering each cell line response as a replicate or applying any sort of merging strategy to dilute cell line specific responses. In addition a short discussion on this aspect might be included, i.e. how cell line from different tissues and with different somatic mutations respond differently at the transcriptional level to the same compound.
- * at the beginning of the Results the authors claim 'Individually, no major anti-HT drug was found to increase ACE2 expression'. It is not clear what 'major' refers to. Does this refer to 'widely used' or 'widely prescribed' ?
- * The final co-expression analysis builds on the assumption that ACEIs down-regulate the expression of their targets. At what extent this is true? there are several examples of drugs upregulating the expression of the genes coding for their targeted proteins (PMID: 20838579). This should be discussed.
- * A description of how the MoA enrichment analysis, whose results are presented in figure 1BD is totally missing. How the individual drugs were aggregated into classes? additionally, how the GSEA was run in this case? intuitively, by sorting drugs based on their effect on ACE2 and then using drug classes as GSEA 'signatures' ? this must be necessarily detailed in the methods.
- * A final figure (or figure panel) summarising the result from the in vivo confirmatory analysis should be included and most promising confirmed hits clearly stated and discussed

Minors

- * There are several undefined acronyms in figure 1BD, such as CCB, ARB, ACEi. Although these are defined in the main, they should be explicated in the figure or the legend.
- * what is the sorting criteria for the classes of drugs in the x-axis labels of figure 1BD? wouldn't make more sense to increase readability via sorting them based on effect and magnitude of ACE

differential expression?

Reviewer #3:

The paper "Systematic cell line-based identification of drugs modifying ACE2 expression" by Sinha et al." reports results from mining public databases for available in vitro and in vivo expression data of the ACE2 gene to identify the effects of clinically approved drugs. Panobinostat and isotretinoin are found to be the top ACE2 up- and down-regulators, respectively. The authors suggest that their results provide specific leads guiding further studies aimed at carefully assessing specific drug effects on ACE2 expression.

The key motivation for studying drug effects on ACE2 at the moment is that it the key host receptor for SARS-CoV-2 entry, and that ACE2 inhibitors are used to treat hypertension and diabetes, which are known to increase risk of a serious or fatal Covid-19 disease course. In fact, a recent study (<https://doi.org/10.1016/j.cca.2020.03.031>) provided evidence that ACE polymorphisms known to be associated with ACE2 expression also correlate with the number of cases per million inhabitants across 25 European countries. This work may be important, since so far "solid evidence concerning whether ACE2 expression can alter the risk of COVID-19 infection is lacking", as the authors say themselves. Even more so, the authors also acknowledge "the role of ACEIs/ARBs in modulating the clinical course of viral pneumonia and COVID-19 infection is also under debate [16-18], both being beyond the scope of this study".

The impact of this paper therefore hinges on these aspects to be studied and clarified, an effort that is surely taken up by many at the moment. I would therefore agree with the authors' claim that there is a "need to investigate the effect on modulating ACE2 expression of a variety of prescribed drugs."

Amongst the cell lines available in CMAP the authors focus on epithelial cancer cell lines, arguing that they likely best resemble airway epithelial cells in the lung, which are implicated in the Covid-19 caused pneumonia. They also mined GEO and GTEx expression data from human lung tissue to check for concordance in the much smaller subset of 10 drugs for which there was in vivo gene expression data. The tissue or cell-type specific aspects of ACE2 expression are indeed important, and the argument to focus on epithelial cells makes sense to me. Yet, I found it surprising that the ACE inhibitors (ACEI) only as a group, but not individually, resulted in a significant ACE2 up-regulation (while this is their known mechanism of action to reduce hypertension). I would therefore recommend checking if their effect is stronger in cells or cell-types directly relevant for the angiotensin induced vaso-constrictive action modulating blood pressure (such as those derived from vascular smooth muscle cells). This could give an idea to what extent the in vitro effects of ACE2 expression changes captured in CMAP cultures can be used as a model for in vivo effects.

The authors noted that none of the six drugs currently being investigated in clinical trials for treating COVID-19 show any significant alteration of ACE2 expression. While interesting, this is probably expected for the antiviral drugs Lopinavir and Ritonavir, which act as protease inhibitors. Amongst these drugs Losartan is the only one known to act as ARB (which also were found as a drug class to not alter ACE2 expression).

Generally, the paper is well written and the analyses look sound to me. I do not think that the paper provides any methodological advancement; it just applies standard analyses to a gene that is of high relevance at the moment. A lot of the significant results apply to drug groups, with some

notable exceptions, such as tomosifan and crizotinib, and the three drugs mentioned in the abstract. While the drug group findings are interesting, it is only the latter that provide "leads for further in vitro and in vivo studies", so maybe more data should be shown for those drugs (e.g. different exposure times, see comment below).

Major comments:

Since this study focuses on ACE2 I wonder if there should not be a bit more information on this gene and its role both in regulating blood pressure and as viral entry point?

Also, it would be good to have more information on CMAP. For example, many experiments there are done using the L1000 high-throughput gene expression assay, which only assesses close to 1000 genes directly and imputes expression estimates for ~11k other genes (with a reduction to 80% in found relationships). Is ACE2 directly assessed by L1000? If not, how well is it imputed?

Time measurements of drug exposure were taken after 24h. Patients on HT treatment are usually roughly at steady stage of exposure, so their ACE2 expression is not in response to exposure onset, but constant exposure. Therefore if measurements taken after longer exposure than 24h are available, this could be more relevant. I suggest checking that and if available confirm that the results are robust with respect to exposure time. (Also for some of the highlighted individual drugs it would be good to know if they are metabolized by the cells in the assay or basically continue acting at the initial concentrations.)

Minor comments:

Fig. 1BDE: Maybe order classes by significance from top to bottom?

It would be great if the authors could publish their analysis code along with the paper, as this would allow others to expand or modify their approach and apply it to different data. Also, it's just good practise for reproducible science.

Reviewer# 1:

The study on ACE2 expression by a wide variety of clinically approved drugs is an important contribution to better understand the consequences such drugs might have for COVID-19 patients. However, there are some parts of the study that need to be explained in more detail. The use of 4 cancer cell lines to test the effect of drugs on ACE2 expression seems not the best approach, as cancer cells are inhibited by ACE2. It would have been better to use some of the non-tumour bronchial epithelial cell lines, but may be the authors can provide a bit more information why they had chosen these cells.

Overall, the study has merits to be published after minor corrections.

We thank reviewer 1 for the feedback and constructive suggestions. In particular, we agree that the use of 4 cancer cell lines was not ideal. The choice was partly due to limited data availability and our initial aim to test all drugs on a shared, common subset of cell-lines, as explained below and in the revised main text (**Page 3-4**). Nevertheless, following up on the reviewer's suggestion, we have tried to the best of our capacity to expand our analysis to additional, potentially more relevant cell types. Please refer to our specific reply to your Comment #3 below.

1. The correspondence by Fang et al hypothesized that hypertension and diabetes develop severe COVID-19 because the two diseases have been reported to express ACE2 at an increased level. In addition, these patients are often treated with drugs that further increase ACE2. Please correct the statement.

We thank the reviewer for pointing out this error. The correction has been made (**Pages 2**):

A recent publication by Fang et al. has suggested that patients with hypertension (HT) and diabetes mellitus may be at higher risk of having severe COVID-19 disease (Fang et al, 2020), as these patients have been reported to express ACE2 at an increased level and are often treated with ACE inhibitors (ACEIs) or angiotensin II type-I receptor blockers (ARBs), which have been previously suggested to increase ACE2 expression

2. Page 2: References # 3 is incorrectly cited. They did not provide data showing that any of the drugs increased ACE2. Replace by e.g. Liang Y, et al The use of cancer cell Kidney Blood Press Res. 2015;40(2):101-10. doi: 10.1159/000368486. Epub 2015 Mar 13; or by Vuille-dit-Bille RN, et al. Amino Acids. 2015 Apr;47(4):693-705. doi: 10.1007/s00726-014-1889-6.

We thank the reviewer for pointing out this error in the references. We replaced the incorrect reference with both references as suggested.

3. Why did the authors use four cancer cell lines to determine the effect of antihypertensive drugs on ACE2. It would have been advisable to use cell lines such as: BEAS2 or NuLi or NCI

H441 for these experiments. There are also a number of commercial available primary human bronchial epithelial cells available. The problem with this approach is that ACE2 is inhibiting many epithelium derived cancer types (Cheng Q, et al. Oncol Rep. 2016 Sep;36(3):1403-10. doi: 10.3892/or.2016.4967; Zong H, et al. Tumour Biol. 2015 Jul;36(7):5171-7. doi: 10.1007/s13277-015-3171-2; Yu X, et al. Int J Mol Med. 2018 Jan;41(1):409-420. doi: 10.3892/ijmm.2017.3252.)

First, we agree with the reviewer that the use of four cancer cell lines was not ideal. In that part of our study, we relied on the analysis of the CMAP dataset, where most of its data is in cancer cell lines and no primary bronchial epithelial cells, including the ones mentioned by the reviewer, are available. Although two non-cancerous lung cells were listed in the CMAP document from https://clue.io/connectopedia/core_cmap_cell_panel, these two cell types are actually not included in their transcriptome profiling experiments. We focused on carcinoma cell lines due to their epithelial origin, and thus they may be more similar to the airway epithelium (likely one of the primary points of viral infection) compared to some of the other cell types (eg. leukemia, sarcoma or lymphoma cells) in CMAP. We have changed the main text to make this point clearer (**Pages 3-4**). The specific choice of these four carcinoma cell lines is again due to the practical issue of data availability and sample size -- a reasonably large proportion of clinically approved drugs are tested on all of these four cell lines, whereas choosing other sets of cell lines or including more cell lines will result in a large variety and inhomogeneity in the identify of the cell-lines that each drug has been tested on, which may act as a confounding factor in the analysis. We have now also mentioned this point in the main text (**Pages 3-4**) and added an Appendix Note and figure (**Appendix Note 1, Figure S1**) showing the trade-off between the number of cell lines to include and the number of drugs tested.

The pertaining text on **Pages 3-4** reads:

Among the available cell types from CMAP, we focused on carcinoma cell lines, since they are of epithelial origin and may bear more resemblance to airway epithelium, a major site of viral entry. We identified 48 clinically approved anti-HT drugs that were tested on the same four carcinoma cell lines for 24 hours in CMAP, and computed the drug-induced ACE2 expression changes averaged across the cell lines (Methods; the cell lines are A549, MCF7, PC3 and VCAP, selected because of the data available for a high number of drugs tested on all these cells, see Appendix and Table EVIA for details).

Nevertheless, we have tried our best to expand our analysis to additional, potentially more relevant sample types. First, we analyzed the data in several normal or primary cells from CMAP, although they represent a much smaller proportion of data covering fewer drugs. These include the HA1E normal kidney cell line, the PHH primary liver cell, and the neural progenitor cell (NPC) differentiated from fibroblast-derived iPSC. These cell/tissue types can also be affected by SARS-CoV-2 (Zaim et al, 2020). We found that although these cells have diverse ACE2 expression responses to drugs, there are some consistent patterns from the drug class enrichment analysis; these analyses have now been added to Results (**Page 6**). The new text summarizing these results read as follows:

The various analyses described above were performed by aggregating the drug-induced expression changes across cell types (as explained above; Methods). We next analyzed CMAP data of additional relevant cell types separately by their tissue of origin to investigate the potential tissue-specific effects. We focused on the lung, kidney, liver, central nervous system (CNS), and intestine (Methods), which represent tissues that can be affected by SARS-CoV-2 (Zaim et al, 2020). For each of these tissues, we were only able to find one (or two, for lung) cell types where a reasonable number (>100) of clinically approved drugs were tested (at the 24 hours time point; details in Table EV2D). However, the cells identified from kidney, liver and CNS were non-cancerous or primary cells (HA1E, PHH, and NPC cells, respectively), which can be more relevant for our investigation. As expected, the drug-induced ACE2 expression changes exhibit mostly weak correlations across cells from different tissue types, with Spearman's correlation coefficients between the log fold-changes of pairs of cells ranging from -0.07 to 0.2 (Figure EV1A; Table EV2E). Nevertheless, we observed a consistent but insignificant trend that ACEIs tend to upregulate ACE2 expression across the three normal cell types from kidney, liver, and CNS (Figure EV1B). Concordant with the findings from the four carcinoma cell lines above, antineoplastic agents as a group were found to be enriched for drugs down-regulating ACE2 in the normal NPC cells from CNS (GSEA adjusted $P=0.01$, Figure EV1C, Table EV2F).

Second, we expanded our previous search and analysis of datasets from the GEO database to include many more datasets on non-cancerous lung and kidney cells, including primary human bronchial epithelial BEAS-2B cells and in vivo lung/kidney samples from humans or rodents. We note that although these datasets cover more relevant cell/tissue types, the number of covered drugs was limited. From this analysis, we found additional significant ACE2 modulating drugs that we highlighted and discussed in an extended paragraph (**Pages 7-8**) and in the new **Figure 2**. The new text and the revised Figure are enclosed below, for convenience and completeness:

To further extend our analysis beyond the CMAP dataset, we mined the GEO database for gene expression data of drug treatments with matched controls in lung and kidney tissue or cells (Methods). We collected a total of 74 relevant lung datasets involving 42 unique clinically approved drugs, among which 27 datasets (covering 21 drugs) were composed of non-cancerous samples including primary bronchial epithelial cells and in vivo samples from human and rodents (Table EV3A). Similarly, for kidney, 35 datasets for 29 drugs (including 23 drugs in 28 non-cancer datasets involving in vivo samples) were identified (Table EV3B). The drug-induced ACE2 differential expression results (Methods) for the lung and kidney datasets are summarized in Figure 2A and 2C, respectively. The results for the top significant drugs identified from the more relevant non-cancer datasets are highlighted in Figure 2B and 2D.

For lung, the top significant drug is dexamethasone, which upregulates ACE2 in datasets of both normal and Pneumocystis-infected mice lung tissue (logFC=0.97 and 0.36, adjusted P=0.001 and 0.027, respectively, Figure 2B). Consistently, dexamethasone also increased ACE2 expression in our analysis of four carcinoma cell lines from the CMAP dataset (logFC=0.18, P=0.006, Table EV1F). Interestingly, corticosteroids, including dexamethasone, have been widely used for severe acute respiratory syndrome (SARS) but showed no survival benefit and possible harm (Russell et al, 2020), and WHO does not recommend the routine use of corticosteroids for COVID-19 patients (World Health Organization, 2020). Another top identified drug is the epidermal growth factor receptor (EGFR) inhibitor erlotinib, which is found to upregulate ACE2 in a dataset of human primary bronchial epithelial cells (logFC=1.04, adjusted P=2.95E-5, Figure 2B), a relevant cell type suggested to interact with the SARS-CoV-2 virus (Mason et al, 2020). In the CMAP analysis, we observed a non-significant trend of ACE2 upregulation at 24 hours by erlotinib (logFC=0.05, adjusted P>0.1, Table EV1F). Interestingly, erlotinib has actually been reported to inhibit the endocytosis and intracellular trafficking of multiple viruses including hepatitis C, dengue and Ebola, exerting broad-spectrum antiviral effects (Bekerman et al, 2017). The chemotherapeutic drug bleomycin is a significant ACE2 down-regulator identified in a dataset of rat lung tissue (logFC=-0.17, adjusted P=0.003, Figure 2B), in accordance with an earlier reports that it decreases ACE2 protein level in alveolar epithelial cells (Uhal et al, 2010).

Among the top significant candidates arising from the kidney cells analysis (summarized in Figure 2C), we again focused on non-cancer datasets and observed that the chemotherapy drug cisplatin up-regulated ACE2 in mice kidney samples while it down-regulated ACE2 in the renal cortex of rat (logFC=0.29 and -1.16, adjusted P=8.06E-3 and 2.36E-5, respectively, Figure 2D), suggesting a cell type and possibly species specific effect. Vancomycin, another top identified drug, is a glycopeptide antibiotic that increases ACE2 expression in mice kidney samples from two independent datasets (logFC=0.89 and 0.93, adjusted P=0.04 for both). Glycopeptide antibiotics and its derivatives have been previously shown to block MERS and SARS cell entry (Zhou et al, 2016). Probenecid, a drug for treating gout, was found to decrease ACE2 expression in a renal cortical cell line (logFC=-0.61, adjusted P=0.001, Figure 2D); this drug has been proposed to be repurposed for anti-influenza therapy (Perwitasari et al, 2013). Other top significant drugs arising from the analysis of cancer datasets of lung and kidney from GEO are shown in Figure EV3 with additional information in Appendix Note 2.

A Lung in vivo & in vitro from GEO

B

C Kidney in vivo & in vitro from GEO

D

Figure 2: Drug-induced ACE2 differential expression in lung and kidney-derived cells or tissue types from the GEO database. This figure summarizes the differential expression analysis results for ACE2 upon treatment by clinically

approved drugs in samples of lung and kidney datasets mined from the GEO database, spanning both cancer and non-cancer datasets (Methods). Volcano plots showing the log fold-change (X-axis) and uncorrected negative log₁₀P value (Y-axis) of ACE2 expression changes are displayed in (A) for 74 datasets of cells or tissue samples from lung, involving 42 clinically approved drugs, and in (C) for 35 kidney datasets involving 29 clinically approved drugs. Among these, we focused on the top significant drugs that modulate ACE2 expression in non-cancerous cells or tissue samples from lung and kidney, which are shown in (B) and (D), respectively, where the ACE2 expression (Y-axis) difference between control and treated groups (X-axis) are shown with boxplots. In the title of each boxplot, the GEO identifier of the respective studies and the corresponding drug name and the sample type are provided. Different datasets involving experiments using the same drug were analyzed and presented separately, since the sample types can be different across datasets. All the P values are computed from differential expression analysis using limma (Ritchie et al, 2015) (Methods). Abbreviations: HBEC, human primary bronchial epithelial cells.

4. On page 6 the authors state that the drugs effect on ACE2 expression was verified 28 cell lines. Please provide more information on the cell type and of their origin. It is unclear if these 28 cell lines result from the different studies or if the authors were performing these assays. If the latter is true, than they should provide more details of the cells.

The data for the 28 cell types analyzed here are also from the CMAP dataset. In the analyses from the previous parts of our study where we used only 4 carcinoma cell lines, we required that all drugs be tested on all included cell lines. In the new analysis, which is **now on Page 5**, we extended the total number of cell types by allowing each drug to be tested on a different set of cells. We added explanations in the Methods section under the subsection “*The CMAP data*” (**Pages 9-10**). The details of these 28 cell lines are given in **Table EV2A**. The text added reads as follows:

***Page 5:** The 28 cell lines include 16 cancer and 12 normal cells; this list is provided in table EV2A with details including primary site, subtype and donor demographics) where each drug may have been tested on a different subset of cells (Methods).*

***Page 10:** In the later parts of our analyses, (1) trading biological homogeneity for higher coverage, we included drugs that each can be tested on a different number and types of cells, resulting in 989 clinically approved drugs tested on a total of 28 cell types, which we used for the analysis of enrichment of drug classes by WHO ATC indications.*

Reviewer #2:

The authors present results from a quick and simple computational analysis of public available repositories of transcriptional data to the aim of identifying modulators of the expression of ACE2: a receptor required for SARS-CoV-2 infection in humans.

To this aim, the authors queried the connectivity map (cMap) database of transcriptional responses to drug treatment in immortalised human cancer cell lines, initially focusing on anti-hypertensive and anti-diabetic drugs, i.e. ACE inhibitors (ACEi) and angiotensin II type I receptor blockers (ARBs). Consistently with previous findings (PMID: 20838579) the authors found that ACE inhibitors increases ACE2 expression but no ARBs was associated with differential expression of ACE2.

Furthermore the authors extended this analysis to all approved compounds with data available in the cMap and performed an enrichment analysis of drug Mode-of-action and therapeutic applications among the compounds exerting an effect on ACE2 expression.

Finally, the authors confirmed some of their hits using an independent dataset including drug transcriptional responses in vivo and attempted confirmatory a gene co-expression analysis using public data from normal lung tissue.

The recent COVID-19 outbreak and the need for findings and results that might help discovering/repositioning drugs able to reduce infections rate and symptoms make the subject of this report timely and important. However the analytical approach presented in this report is not novel, there is no new data presented nor final strong statements or guidelines for experimental followups, with just percentages of drugs with confirmatory in-vivo results reported, and not even stating the names of the most promising hits.

In addition, the first part of the paper focuses too much on confirmatory results and previously reported findings related to ant hypertensive and anti-diabetic drugs, whereas the more interesting unbiased and comprehensive analysis of all approved compounds is presented quite poorly.

We thank the reviewer for the helpful feedback and constructive criticism of our manuscript. Accordingly we have revised the parts of our manuscript on the analysis of all clinically approved drugs by clarifying the methods used (**addressing the reviewer's Comment #1 and #4**) and included clear descriptions of the top drugs identified. A panel was added to Figure 1 (current **Figure 1D**) that explicitly shows the control vs drug-treated *ACE2* expression for the top significant drugs across all clinically approved drugs. We have also expanded our analysis of datasets from the GEO database, identifying additional clinically approved drugs that significantly alter *ACE2* expression in relevant cell/tissue types. We have summarized these results in a new main text **Figure 2** and added the corresponding description in the Results (**Page 9-10, addressing the Comment #5**). Please see the reply to specific comments below.

Particularly the following points should be addressed, in my opinion:

1. The methods section should be generally extended, for example is not clear whether the differential ACE2 expression analysis for a given drug was performed considering each cell line response as a replicate or applying any sort of merging strategy to dilute cell line specific responses. In addition a short discussion on this aspect might be included, i.e. how cell line from different tissues and with different somatic mutations respond differently at the transcriptional level to the same compound.

We have comprehensively expanded the Methods to include separate subsections describing the CMAP data, the differential expression analysis and the drug class enrichment analysis. In particular, we have clarified the details on how the differential expression analysis was performed for multiple cell lines. In the analysis of the four carcinoma cell lines from CMAP, we controlled for cell line identity as a covariate in the limma linear model, and therefore the results represent averaged differential expression across the four cell lines. We have made this point clearer both in the Methods (subsection titled “*Identification of ACE2 modulators from the CMAP dataset*”) and Results (**Page 4**); the text on Page 4 now reads:

We identified 48 clinically approved anti-HT drugs that were tested on the same four carcinoma cell lines for 24 hours in CMAP, and computed the drug-induced ACE2 expression changes averaged across the cell lines (Methods; the cell lines are A549, MCF7, PC3 and VCAP, selected because of the data available for a high number of drugs tested on all these cells, see Appendix and Table EV1A for details).

Indeed cell lines with different cells of origin or genomic background may respond differently (PMID: 26824188, 28071740, 31990955, <https://doi.org/10.1101/868752>). We have extended our analysis of ACE2 expression changes in response to drugs to several normal cell types of several tissues of origin that may be affected by COVID-19 (including lung, kidney, liver, the central nervous system, and intestine, each tissue type analyzed separately). In general, we find that the concordance of ACE2 differential expression profiles across cell types are low (Figure EV1A), however, some similarities among the enriched drug classes could be observed (Figure EV1B,C). These results are now described in the Results (**Page 6**) and read as follows:

The various analyses described above were performed by aggregating the drug-induced expression changes across cell types (as explained above; Methods). We next analyzed CMAP data of additional relevant cell types separately by their tissue of origin to investigate the potential tissue-specific effects. We focused on the lung, kidney, liver, central nervous system (CNS), and intestine (Methods), which represent tissues that can be affected by SARS-CoV-2 (Zaim et al, 2020). For each of these tissues, we were only able to find one (or two, for lung) cell types where a reasonable number (>100) of clinically approved drugs were tested

(at the 24 hours time point; details in Table EV2D). However, the cells identified from kidney, liver and CNS were non-cancerous or primary cells (HA1E, PHH, and NPC cells, respectively), which can be more relevant for our investigation. As expected, the drug-induced ACE2 expression changes exhibit mostly weak correlations across cells from different tissue types, with Spearman's correlation coefficients between the log fold-changes of pairs of cells ranging from -0.07 to 0.2 (Figure EV1A; Table EV2E). Nevertheless, we observed a consistent but insignificant trend that ACEIs tend to upregulate ACE2 expression across the three normal cell types from kidney, liver, and CNS (Figure EV1B). Concordant with the findings from the four carcinoma cell lines above, antineoplastic agents as a group were found to be enriched for drugs down-regulating ACE2 in the normal NPC cells from CNS (GSEA adjusted P=0.01, Figure EV1C, Table EV2F).

We also added a brief summary in the Discussion section (**Page 13**):

Analyzing the additional but limited amount of CMAP data for cells from different tissue types and for drug treatment for longer durations up to 48 hours, we find that the drug-induced ACE2 changes can be tissue-specific and time-dependent (Figure EV1,2), and for some drugs, their effect on ACE2 can become stronger upon prolonged treatment (Figure EV2).

2. at the beginning of the Results the authors claim 'Individually, no major anti-HT drug was found to increase ACE2 expression'. It is not clear what 'major' refers to. Does this refer to 'widely used' or 'widely prescribed' ?

Yes this is meant to be “widely prescribed”, including some members of the ACEI, ARB, and calcium channel blockers. The text has been modified considering this comment (**Page 4**):

*“Individually, no **widely prescribed** anti-HT drug was found to increase ACE2 expression significantly in these experiments, except for methyldopa (an alpha-2 adrenergic receptor agonist) and molsidomine (a vasodilator) do significantly decrease ACE2 expression”*

3. The final co-expression analysis builds on the assumption that ACEIs down-regulate the expression of their targets. At what extent this is true? there are several examples of drugs upregulating the expression of the genes coding for their targeted proteins (PMID: 20838579). This should be discussed.

One underlying assumption of this analysis is actually that the functional effects induced by a drug (which usually acts via inhibiting its target protein) tend to be similar to the effects of down-regulating the drug's target on the mRNA level. We note that this is different from the assumption that the drugs will downregulate the expression of their targets, the latter being a stronger assumption that is indeed less likely to be true in general and thank the reviewer for

noting that. Nevertheless, we still think this additional analysis is valuable and chose to maintain it, but have added a clarifying sentence to this extent on (Page 12), which reads:

This analysis is based on the notion that the effect of drug treatment, which usually acts on the protein level, is in general functionally similar to the effect of downregulating the drug target expression on the mRNA level.

4. A description of how the MoA enrichment analysis, whose results are presented in figure 1BD is totally missing. How the individual drugs were aggregated into classes? additionally, how the GSEA was run in this case? intuitively, by sorting drugs based on their effect on ACE2 and then using drug classes as GSEA 'signatures' ? this must be necessarily detailed in the methods.

We thank the reviewer for pointing this out. We sincerely apologize for the missing information, which we have of course rectified now. We have expanded the description of the GSEA analysis into a separate section titled “Analysis of drug classes enrichment in ACE2 modulators” in the Methods section (Page 15). Basically, the understanding of the reviewer is correct -- all the 672 clinically approved drugs were sorted by their induced ACE2 expression log fold-changes, then GSEA was applied by using the classes of drugs by MOA as “gene sets” (here actually “drug sets”). The annotation on classes of drugs by MOA was obtained from the Drug Repurposing Hub (Corsello et al, 2017). The corresponding new text now reads:

Specifically, the drug-induced ACE2 expression log fold-change values were ranked, and the GSEA method was applied to the ranked list with “gene sets” for GSEA being the sets of drugs of each class. The drug classes were based on mechanisms of action (MOA) and indication in our two respective analysis described in the main text, where the MOA annotation was obtained from the Drug Repurposing Hub (Corsello et al, 2017), and for drug indication we used the WHO ATC classification (World Health Organization, 2006) obtained from the DrugBank database (Wishart et al, 2018).

5. A final figure (or figure panel) summarising the result from the in vivo confirmatory analysis should be included and most promising confirmed hits clearly stated and discussed

Thanks. First, we updated the curation and analysis of GEO datasets to include more in vivo datasets and relevant in vitro datasets such as those in primary airway epithelial cells. In addition to the lung, we also collected data from the kidney, another tissue site that can be affected by COVID-19. Second, following the reviewer’s suggestion we have added a new figure (Figure 2) summarizing the results from analyzing these datasets, where the top identified drugs that can alter ACE2 expression were also shown. We have updated the description of this analysis and included an extensive description of the top drugs in the Results (Page 7-8). The new text is enclosed below, including the new Figure, as follows:

To further extend our analysis beyond the CMAP dataset, we mined the GEO database for gene expression data of drug treatments with matched controls in lung and kidney tissue or cells (Methods). We collected a total of 74 relevant lung datasets involving 42 unique clinically approved drugs, among which 27 datasets (covering 21 drugs) were composed of non-cancerous samples including primary bronchial epithelial cells and in vivo samples from human and rodents (Table EV3A). Similarly, for kidney, 35 datasets for 29 drugs (including 23 drugs in 28 non-cancer datasets involving in vivo samples) were identified (Table EV3B). The drug-induced ACE2 differential expression results (Methods) for the lung and kidney datasets are summarized in Figure 2A and 2C, respectively. The results for the top significant drugs identified from the more relevant non-cancer datasets are highlighted in Figure 2B and 2D.

For lung, the top significant drug is dexamethasone, which upregulates ACE2 in datasets of both normal and Pneumocystis-infected mice lung tissue ($\log_{2}FC=0.97$ and 0.36 , adjusted $P=0.001$ and 0.027 , respectively, Figure 2B). Consistently, dexamethasone also increased ACE2 expression in our analysis of four carcinoma cell lines from the CMAP dataset ($\log_{2}FC=0.18$, $P=0.006$, Table EV1F). Interestingly, corticosteroids, including dexamethasone, have been widely used for severe acute respiratory syndrome (SARS) but showed no survival benefit and possible harm (Russell et al, 2020), and WHO does not recommend the routine use of corticosteroids for COVID-19 patients (World Health Organization, 2020). Another top identified drug is the epidermal growth factor receptor (EGFR) inhibitor erlotinib, which is found to upregulate ACE2 in a dataset of human primary bronchial epithelial cells ($\log_{2}FC=1.04$, adjusted $P=2.95E-5$, Figure 2B), a relevant cell type suggested to interact with the SARS-CoV-2 virus (Mason et al, 2020). In the CMAP analysis, we observed a non-significant trend of ACE2 upregulation at 24 hours by erlotinib ($\log_{2}FC=0.05$, adjusted $P>0.1$, Table EV1F). Interestingly, erlotinib has actually been reported to inhibit the endocytosis and intracellular trafficking of multiple viruses including hepatitis C, dengue and Ebola, exerting broad-spectrum antiviral effects (Bekerman et al, 2017). The chemotherapeutic drug bleomycin is a significant ACE2 down-regulator identified in a dataset of rat lung tissue ($\log_{2}FC=-0.17$, adjusted $P=0.003$, Figure 2B), in accordance with an earlier reports that it decreases ACE2 protein level in alveolar epithelial cells (Uhal et al, 2010).

Among the top significant candidates arising from the kidney cells analysis (summarized in Figure 2C), we again focused on non-cancer datasets and observed that the chemotherapy drug cisplatin up-regulated ACE2 in mice kidney samples while it down-regulated ACE2 in the renal cortex of rat ($\log_{2}FC=0.29$ and -1.16 , adjusted $P=8.06E-3$ and $2.36E-5$, respectively, Figure 2D), suggesting a cell type and possibly species specific effect. Vancomycin, another top identified drug, is a glycopeptide antibiotic that increases ACE2 expression in mice kidney samples from two independent datasets ($\log_{2}FC=0.89$ and 0.93 , adjusted $P=0.04$

for both). Glycopeptide antibiotics and its derivatives have been previously shown to block MERS and SARS cell entry (Zhou et al, 2016). Probenecid, a drug for treating gout, was found to decrease ACE2 expression in a renal cortical cell line (logFC=-0.61, adjusted P=0.001, Figure 2D); this drug has been proposed to be repurposed for anti-influenza therapy (Perwitasari et al, 2013). Other top significant drugs arising from the analysis of cancer datasets of lung and kidney from GEO are shown in Figure EV3 with additional information in Appendix Note 2.

A Lung in vivo & in vitro from GEO

B

C Kidney in vivo & in vitro from GEO

D

Figure 2: Drug-induced ACE2 differential expression in lung and kidney-derived cells or tissue types from the GEO database. This figure summarizes the differential expression analysis results for ACE2 upon treatment by clinically

approved drugs in samples of lung and kidney datasets mined from the GEO database, spanning both cancer and non-cancer datasets (Methods). Volcano plots showing the log fold-change (X-axis) and uncorrected negative log₁₀P value (Y-axis) of ACE2 expression changes are displayed in (A) for 74 datasets of cells or tissue samples from lung, involving 42 clinically approved drugs, and in (C) for 35 kidney datasets involving 29 clinically approved drugs. Among these, we focused on the top significant drugs that modulate ACE2 expression in non-cancerous cells or tissue samples from lung and kidney, which are shown in (B) and (D), respectively, where the ACE2 expression (Y-axis) difference between control and treated groups (X-axis) are shown with boxplots. In the title of each boxplot, the GEO identifier of the respective studies and the corresponding drug name and the sample type are provided. Different datasets involving experiments using the same drug were analyzed and presented separately, since the sample types can be different across datasets. All the P values are computed from differential expression analysis using limma (Ritchie et al, 2015) (Methods). Abbreviations: HBEC, human primary bronchial epithelial cells.

Minors

6. There are several undefined acronyms in figure 1BD, such as CCB, ARB, ACEi. Although these are defined in the main, they should be explicated in the figure or the legend.

Explanations of these acronyms have been added to the Figure 1 legend.

7. what is the sorting criteria for the classes of drugs in the x-axis labels of figure 1BD? wouldn't make more sense to increase readability via sorting them based on effect and magnitude of ACE differential expression?

Thanks. The previous sorting was performed alphabetically. We agree with the reviewer that sorting based on enrichment significance may improve the interpretability and thus have modified Figure 1 accordingly. We preferred to sort by P-value rather than by effect size. (Reviewer 3, minor comment 4 also recommended sorting by P-values)

Reviewer #3:

The paper "Systematic cell line-based identification of drugs modifying ACE2 expression" by Sinha et al." reports results from mining public databases for available in vitro and in vivo expression data of the ACE2 gene to identify the effects of clinically approved drugs. Panobinostat and isotretinoin are found to be the top ACE2 up- and down-regulators, respectively. The authors suggest that their results provide specific leads guiding further studies aimed at carefully assessing specific drug effects on ACE2 expression.

The key motivation for studying drug effects on ACE2 at the moment is that it the key host receptor for SARS-CoV-2 entry, and that ACE2 inhibitors are used to treat hypertension and diabetes, which are known to increase risk of a serious or fatal Covid-19 disease course. In fact, a recent study (<https://doi.org/10.1016/j.cca.2020.03.031>) provided evidence that ACE polymorphisms known to be associated with ACE2 expression also correlate with the number of cases per million inhabitants across 25 European countries. This work may be important, since so far "solid evidence concerning whether ACE2 expression can alter the risk of COVID-19 infection is lacking", as the authors say themselves. Even more so, the authors also acknowledge "the role of ACEIs/ARBs in modulating the clinical course of viral pneumonia and COVID-19 infection is also under debate [16-18], both being beyond the scope of this study".

The impact of this paper therefore hinges on these aspects to be studied and clarified, an effort that is surely taken up by many at the moment. I would therefore agree with the authors' claim that there is a "need to investigate the effect on modulating ACE2 expression of a variety of prescribed drugs."

Amongst the cell lines available in CMAP the authors focus on epithelial cancer cell lines, arguing that they likely best resemble airway epithelial cells in the lung, which are implicated in the Covid-19 caused pneumonia. They also mined GEO and GTEx expression data from human lung tissue to check for concordance in the much smaller subset of 10 drugs for which there was in vivo gene expression data. The tissue or cell-type specific aspects of ACE2 expression are indeed important, and the argument to focus on epithelial cells makes sense to me. Yet, I found it surprising that the ACE inhibitors (ACEI) only as a group, but not individually, resulted in a significant ACE2 up-regulation (while this is their known mechanism of action to reduce hypertension). I would therefore recommend checking if their effect is stronger in cells or cell-types directly relevant for the angiotensin induced vaso-constrictive action modulating blood pressure (such as those derived from vascular smooth muscle cells). This could give an idea to what extent the in vitro effects of ACE2 expression changes captured in CMAP cultures can be used as a model for in vivo effects.

Thank you. We agree with the reviewer that investigating the drug-induced effects on *ACE2* in additional relevant cell types can prove informative. We note that ACEIs act via the inhibition of the angiotensin-converting enzyme (ACE) rather than angiotensin-converting enzyme 2 (*ACE2*), which are distinct enzymes within the same pathway but with different functions. ACEIs are not necessarily expected to inhibit (or increase) *ACE2* expression, and actually the effects of ACEIs on the *ACE2* gene expression are largely not well characterized. We have included more background information on the renin-angiotensin pathway, *ACE2* and ACEIs in the Introduction based on the reviewer's Comment #1 below (**Page 3, see Comment #1 below**).

Although overall there are only limited drug-induced gene expression data available, we have tried to the best of our capacity to expand our analysis to additional relevant cell types, i.e. those that represent tissue sites that can be affected by COVID-19. First, we analyzed the data in several normal or primary cells from CMAP, although they represent a much smaller proportion

of data covering fewer drugs (**Page 6**). These include the HA1E normal kidney cell line, the PHH primary liver cell, and the neural progenitor cell (NPC) differentiated from fibroblast-derived iPSC. These cell/tissue types also represent those that can be affected by SARS-CoV-2. Second, we expanded our previous search and analysis of datasets from the GEO database to include many more datasets on non-cancerous lung and kidney samples, including primary human bronchial epithelial BEAS-2B cells and *in vivo* lung/kidney samples from human or rodents. Of note, kidney is also known to be a target organ of ACEIs; we found two *in vivo* kidney datasets for the ACEIs captopril and enalapril, although in both cases the change in *ACE2* is not significant after FDR correction (Table EV3B). The corresponding descriptions in the Results have been updated (**Page 7-8**, see below). The paragraph added on **Page 6** about the analysis of additional cell line in CMAP is attached here:

The various analyses described above were performed by aggregating the drug-induced expression changes across cell types (as explained above; Methods). We next analyzed CMAP data of additional relevant cell types separately by their tissue of origin to investigate the potential tissue-specific effects. We focused on the lung, kidney, liver, central nervous system (CNS), and intestine (Methods), which represent tissues that can be affected by SARS-CoV-2 (Zaim et al, 2020). For each of these tissues, we were only able to find one (or two, for lung) cell types where a reasonable number (>100) of clinically approved drugs were tested (at the 24 hours time point; details in Table EV2D). However, the cells identified from kidney, liver and CNS were non-cancerous or primary cells (HA1E, PHH, and NPC cells, respectively), which can be more relevant for our investigation. As expected, the drug-induced ACE2 expression changes exhibit mostly weak correlations across cells from different tissue types, with Spearman's correlation coefficients between the log fold-changes of pairs of cells ranging from -0.07 to 0.2 (Figure EV1A; Table EV2E). Nevertheless, we observed a consistent but insignificant trend that ACEIs tend to upregulate ACE2 expression across the three normal cell types from kidney, liver, and CNS (Figure EV1B). Concordant with the findings from the four carcinoma cell lines above, antineoplastic agents as a group were found to be enriched for drugs down-regulating ACE2 in the normal NPC cells from CNS (GSEA adjusted $P=0.01$, Figure EV1C, Table EV2F).

The authors noted that none of the six drugs currently being investigated in clinical trials for treating COVID-19 show any significant alteration of ACE2 expression. While interesting, this is probably expected for the antiviral drugs Lopinavir and Ritonavir, which act as protease inhibitors. Amongst these drugs Losartan is the only one known to act as ARB (which also were found as a drug class to not alter ACE2 expression).

Generally, the paper is well written and the analyses look sound to me. I do not think that the paper provides any methodological advancement; it just applies standard analyses to a gene that is of high relevance at the moment. A lot of the significant results apply to drug groups, with

some notable exceptions, such as tomozifan and crizotinib, and the three drugs mentioned in the abstract. While the drug group findings are interesting, it is only the latter that provide "leads for further *in vitro* and *in vivo* studies", so maybe more data should be shown for those drugs (e.g. different exposure times, see comment below).

We thank the reviewer for the constructive comments and overall agree with all the points made here. Indeed, many current drugs under investigation for COVID-19 target different viral proteins and are not expected to alter *ACE2* directly; we examined these drugs and confirmed that they do not significantly modulate *ACE2* expression. On the other hand, we agree that we should provide more detailed results on the top individual *ACE2* modulators identified among all clinically approved drugs. We hence have updated the curation and analysis of GEO datasets to include more *in vivo* and also relevant *in vitro* datasets in both lung and kidney (the latter another tissue site that can be affected by COVID-19). Correspondingly we have added a new figure (**Figure 2**) showing the top significant *ACE2*-modulating drugs identified from these analyses. The description of this analysis and the top candidate drugs arising was also updated in the Results (**Page 7-8**). The pertaining new text and Figure are enclosed below, for convenience and completeness:

To further extend our analysis beyond the CMAP dataset, we mined the GEO database for gene expression data of drug treatments with matched controls in lung and kidney tissue or cells (Methods). We collected a total of 74 relevant lung datasets involving 42 unique clinically approved drugs, among which 27 datasets (covering 21 drugs) were composed of non-cancerous samples including primary bronchial epithelial cells and *in vivo* samples from human and rodents (Table EV3A). Similarly, for kidney, 35 datasets for 29 drugs (including 23 drugs in 28 non-cancer datasets involving *in vivo* samples) were identified (Table EV3B). The drug-induced *ACE2* differential expression results (Methods) for the lung and kidney datasets are summarized in Figure 2A and 2C, respectively. The results for the top significant drugs identified from the more relevant non-cancer datasets are highlighted in Figure 2B and 2D.

For lung, the top significant drug is dexamethasone, which upregulates *ACE2* in datasets of both normal and *Pneumocystis*-infected mice lung tissue ($\log_{2}FC=0.97$ and 0.36 , adjusted $P=0.001$ and 0.027 , respectively, Figure 2B). Consistently, dexamethasone also increased *ACE2* expression in our analysis of four carcinoma cell lines from the CMAP dataset ($\log_{2}FC=0.18$, $P=0.006$, Table EV1F). Interestingly, corticosteroids, including dexamethasone, have been widely used for severe acute respiratory syndrome (SARS) but showed no survival benefit and possible harm (Russell et al, 2020), and WHO does not recommend the routine use of corticosteroids for COVID-19 patients (World Health Organization, 2020). Another top identified drug is the epidermal growth factor receptor (EGFR) inhibitor erlotinib, which is found to upregulate *ACE2* in a dataset of human primary bronchial epithelial cells ($\log_{2}FC=1.04$, adjusted $P=2.95E-5$, Figure 2B), a

relevant cell type suggested to interact with the SARS-CoV-2 virus (Mason et al, 2020). In the CMAP analysis, we observed a non-significant trend of ACE2 upregulation at 24 hours by erlotinib (logFC=0.05, adjusted P>0.1, Table EV1F). Interestingly, erlotinib has actually been reported to inhibit the endocytosis and intracellular trafficking of multiple viruses including hepatitis C, dengue and Ebola, exerting broad-spectrum antiviral effects (Bekerman et al, 2017). The chemotherapeutic drug bleomycin is a significant ACE2 down-regulator identified in a dataset of rat lung tissue (logFC=-0.17, adjusted P=0.003, Figure 2B), in accordance with an earlier reports that it decreases ACE2 protein level in alveolar epithelial cells (Uhal et al, 2010).

Among the top significant candidates arising from the kidney cells analysis (summarized in Figure 2C), we again focused on non-cancer datasets and observed that the chemotherapy drug cisplatin up-regulated ACE2 in mice kidney samples while it down-regulated ACE2 in the renal cortex of rat (logFC=0.29 and -1.16, adjusted P=8.06E-3 and 2.36E-5, respectively, Figure 2D), suggesting a cell type and possibly species specific effect. Vancomycin, another top identified drug, is a glycopeptide antibiotic that increases ACE2 expression in mice kidney samples from two independent datasets (logFC=0.89 and 0.93, adjusted P=0.04 for both). Glycopeptide antibiotics and its derivatives have been previously shown to block MERS and SARS cell entry (Zhou et al, 2016). Probenecid, a drug for treating gout, was found to decrease ACE2 expression in a renal cortical cell line (logFC=-0.61, adjusted P=0.001, Figure 2D); this drug has been proposed to be repurposed for anti-influenza therapy (Perwitasari et al, 2013). Other top significant drugs arising from the analysis of cancer datasets of lung and kidney from GEO are shown in Figure EV3 with additional information in Appendix Note 2.

A Lung in vivo & in vitro from GEO

B

C Kidney in vivo & in vitro from GEO

D

Figure 2: Drug-induced ACE2 differential expression in lung and kidney-derived cells or tissue types from the GEO database. This figure summarizes the differential expression analysis results for ACE2 upon treatment by clinically

approved drugs in samples of lung and kidney datasets mined from the GEO database, spanning both cancer and non-cancer datasets (Methods). Volcano plots showing the log fold-change (X-axis) and uncorrected negative log₁₀P value (Y-axis) of ACE2 expression changes are displayed in (A) for 74 datasets of cells or tissue samples from lung, involving 42 clinically approved drugs, and in (C) for 35 kidney datasets involving 29 clinically approved drugs. Among these, we focused on the top significant drugs that modulate ACE2 expression in non-cancerous cells or tissue samples from lung and kidney, which are shown in (B) and (D), respectively, where the ACE2 expression (Y-axis) difference between control and treated groups (X-axis) are shown with boxplots. In the title of each boxplot, the GEO identifier of the respective studies and the corresponding drug name and the sample type are provided. Different datasets involving experiments using the same drug were analyzed and presented separately, since the sample types can be different across datasets. All the P values are computed from differential expression analysis using limma (Ritchie et al, 2015) (Methods). Abbreviations: HBEC, human primary bronchial epithelial cells.

Major

comments:

1. Since this study focuses on ACE2 I wonder if there should not be a bit more information on this gene and its role both in regulating blood pressure and as viral entry point?

The Introduction of our previous version of manuscript was written in a concise style. In the revised version, following the reviewer's suggestion, we have expanded the Introduction to provide additional background information on the renin-angiotensin pathway, ACE2, and ACEIs (Page 3):

ACEIs and ARBs are widely used antihypertensive drugs acting on the renin-angiotensin system, a hormone system comprising different variants of angiotensin peptides with important roles in regulating vascular and kidney functions (Dimou et al, 2019). ACEIs inhibit the ACE (but not ACE2) gene that converts angiotensin I to angiotensin II, while ARBs suppress the blood pressure-increasing effect of angiotensin II by blocking its binding to its receptor (Dimou et al, 2019). Distinct from ACE, ACE2 is responsible for the conversion of angiotensin I and II into other forms including angiotensin-(1-9) and angiotensin-(1-7), which counteracts the effect of angiotensin II and may paradoxically have protective effects on the lung and on the cardiovascular system (Jiang et al, 2014; Paz Ocaranza et al, 2020).

2. Also, it would be good to have more information on CMAP. For example, many experiments there are done using the L1000 high-throughput gene expression assay, which only assesses close to 1000 genes directly and imputes expression estimates for ~11k other genes (with a reduction to

80% in found relationships). Is ACE2 directly assessed by L1000? If not, how well is it imputed?

Thanks. We have included additional description of the CMAP dataset in Methods as a separate subsection entitled “*The CMAP data*” (Pages 9-10). Specifically, the ACE2 gene is not a directly measured gene (i.e. “landmark gene” according to the CMAP terminology) and its expression levels were imputed using the standard CMAP L1000 protocol described in their publication (Subramanian et al, 2017). However, ACE2 is among the “best-inferred genes” (i.e. “BING” according to the CMAP terminology), which means that during validation of the L1000 protocol against RNA sequencing, the L1000-inferred expression of ACE2 significantly correlated with RNA-seq results with an empirical $P < 0.05$ comparing to null distributions based on correlation with random genes (Subramanian et al, 2017), and therefore the data on ACE2 expression is deemed reasonably reliable. This point is also added to the Methods section titled “*Identification of ACE2 modulators from the CMAP dataset*” (Page 10-11):

Using each of the subsets of CMAP data as described above, we selected the expression data of only the “landmark” and “BING” (best-inferred) genes for the population controls and the drug-treated samples. The landmark/BING genes and population controls are described in the CMAP publication (Subramanian et al, 2017); ACE2 is not a landmark gene but is a best-inferred gene.

3. Time measurements of drug exposure were taken after 24h. Patients on HT treatment are usually roughly at steady stage of exposure, so their ACE2 expression is not in response to exposure onset, but constant exposure. Therefore if measurements taken after longer exposure than 24h are available, this could be more relevant. I suggest checking that and if available confirm that the results are robust with respect to exposure time. (Also for some of the highlighted individual drugs it would be good to know if they are metabolized by the cells in the assay or basically continue acting at the initial concentrations.)

The availability of expression measurements after prolonged treatment of drugs is limited in the CMAP dataset. Nevertheless, following up on the reviewer’s comment, we have now identified and analyzed 14 clinically approved drugs with gene expression measured at 48 hours after treatment from CMAP (none of these are antihypertensive drugs). Among these, we have indeed identified several drugs, including vemurafenib and fluphenazine that achieved higher extent and significance of ACE2 modulation at 48 hours compared to 24 hours, suggesting the effect of some drugs could be stronger with prolonged treatment. We have added these new results (Page 6):

Additionally, we also analyzed CMAP drug-treatment data beyond the 24 hours time point we used above. We identified 14 clinically approved drugs with such data available, which were tested on either the 293T or VCAP cell lines for 48 hours (details and results in Table EV2G). In these data, vemurafenib, fluphenazine, and afatinib were found to significantly upregulate ACE2 expression (adjusted $P < 0.1$), and imatinib significantly down-regulated ACE2

(adjusted $P=0.001$). Six hours and 24 hours treatment data in the same cell line were available for vemurafenib and fluphenazine; for both of these drugs we observed a trend of time-dependent increase in the level of ACE2 upregulation (Figure EV2), suggesting that these drugs may modulate ACE2 expression during prolonged treatment.

We also added a short summary in the Discussion of such time-dependent effects (**Page 9**):

Analyzing the additional but limited amount of CMAP data for cells from different tissue types and for drug treatment for longer durations up to 48 hours, we find that the drug-induced ACE2 changes can be tissue-specific and time-dependent (Figure EV1,2), and for some drugs, their effect on ACE2 can become stronger upon prolonged treatment (Figure EV2).

We note that *in vivo*, drugs are mainly metabolized by the liver and excreted via the kidney, and their blood concentrations typically vary with time after a single dose of drug administration, and typically to achieve prolonged effects multiple doses of drugs need to be given; however, such effects are less relevant in cell line settings, i.e. after the initial drug treatment the drug concentration in the culture media tends to remain largely unchanged for a considerable amount of time, which applies to the CMAP data that we used.

Minor

comments:

4. Fig. 1BDE: Maybe order classes by significance from top to bottom?

Thanks. We have modified the panels of Figure 1 to order the classes by P values. See also our response to reviewer 2, comment 7, which is similar.

5. It would be great if the authors could publish their analysis code along with the paper, as this would allow others to expand or modify their approach and apply it to different data. Also, it's just good practise for reproducible science.

Certainly. We have uploaded all of our codes to a GitHub repository and added the relevant information in the “*Data Availability*” section (**Page 13**).

Thank you for sending us your revised manuscript. We have now heard back from the two reviewers who were asked to evaluate the revised study. As you will see below, both reviewers are satisfied with the modifications made and are supportive of publication. Reviewer #1 only recommends having the text edited by a native English speaker.

We would also ask you to address the following editorial issues.

REFeree REPORTS

Reviewer #1:

The authors addressed all queries. However, the English needs reading and correction by a native speaker.

No further comments

Reviewer #2:

The authors have satisfactorily addressed all mine and other reviewers' points. I recommend their submission for publication in Molecular Systems Biology.

The Authors have made the requested editorial changes.

Thank you again for sending us your revised manuscript. We are now satisfied with the modifications made and I am pleased to inform you that your paper has been accepted for publication.

Corresponding Author Name: Eytan Ruppin

Manuscript Number: MSB-20-9628R